# The association between shift work exposure and cognitive impairment among middle-aged and older adults: Results from the Canadian Longitudinal Study on Aging (CLSA)

**Durdana Khan** ⓘ *, **Heather Edgell** ⓘ, **Michael Rotondi, Hala Tamim**

School of Kinesiology and Health Science, York University, Toronto, Ontario, Canada

* drkhan@yorku.ca

## Abstract

### Background

Shift work, especially rotating and night shift work, has been linked to a wide range of detrimental health outcomes. Occupational factors like shift work and their potential impact on cognitive functions have received little attention, and the evidence is inconclusive. The objective of our study is to explore associations between shift work exposure and cognitive impairment indicators based on comparisons with the normative standards from the Canadian population.

### Methods

Cross-sectional analyses were performed using baseline Canadian Longitudinal Study on Aging database, including 47,811 middle-aged and older adults (45–85 years). Three derived shift work variables were utilized: ever exposed to shift work, shift work exposure in longest job, and shift work exposure in current job. Four cognitive function tests were utilized, Rey Auditory Verbal Learning Tests (immediate and delayed) representing memory domain, and Animal Fluency, and Mental Alteration, representing the executive function domain. All cognitive test scores included in study were normalized and adjusted for the participant's age, sex, education and language of test administration (English and French), which were then compared to normative data to create "cognitive impairment' variables. Unadjusted and adjusted multivariable logistic regression models were used to determine associations between shift work variables and cognitive impairment individually (memory and executive function domains), and also for overall cognitive impairment.

### Result

Overall, one in every five individuals (21%) reported having been exposed to some kind of shift work during their jobs. Exposure to night shift work (both current and longest job) was associated with overall cognitive impairment. In terms of domain-based measures, night shift work (longest job) was associated with memory function impairment, and those

**Data Availability Statement:** Data are available from the Canadian Longitudinal Study on Aging (www.clsa-elcv.ca) for researchers who meet the

criteria for access to de-identified CLSA data. Data access occurs via a data access application to CLSA. Once the data application is approved, data are provided when a data sharing agreement is in place between McMaster University (the custodian of the data) and the researchers' institution, and ethics approval has been submitted to CLSA. Data access is explained at https://www.clsa-elcv.ca/data-access. Applications are submitted to access@clsa-elcv.ca.

**Funding:** Funding for the Canadian Longitudinal Study on Aging (CLSA) is provided by the Government of Canada through the Canadian Institutes of Health Research (CIHR) under grant reference: LSA 94473 and the Canada Foundation for Innovation, as well as the following provinces, Newfoundland, Nova Scotia, Quebec, Ontario, Manitoba, Alberta, and British Columbia.

**Competing interests:** The authors have declared that no competing interests exist.

exposed to rotating shift work (both current and longest job) showed impairment on executive function measures, when compared to daytime workers.

## Conclusion

This study suggests disruption to the circadian rhythm, due to shift work has negative impact on cognitive function in middle-aged and older adults and this warrants further investigation.

## Introduction

Cognitive impairment is a condition that indicates the transitional phase between normal cognitive function and dementia [1]. As the global population is aging, cognitive impairment has become a major health concern affecting independence and quality of life. Although it was believed that cognitive decline was an inevitable feature of the normal aging process [2], evidence now reveals that cognitive functions are modifiable across the lifespan [3]. A comprehensive meta-analysis of 247 studies examined key risk factors such as socio-economic and behavioural factors for cognitive impairment including dementia and found that lower educational attainment and smoking were strong indicators of cognitive disorders [4]. Similarly, lack of availability of social support has been linked to poor cognitive function among middle-aged and older adults [5].

Little attention has been paid to occupational factors like shift work (SW) and their potential role in cognitive performances. In general, SW refers to a work schedule that occurs outside the traditional daytime (9:00 a.m. to 5:00 p.m.) working hours [6–8]. The entire spectrum of SW comprises the following shifts: evening shift, night shift, rotating shift (day to evening and/or night), and other less defined shifts including on-call or casual shift (no set schedule) and irregular shifts [7, 9–12]. A variety of negative health outcomes have been associated with SW, particularly night and rotating SW [9, 10, 13–17]. A meta-analysis of 34 studies showed that both types of SW, night and rotating, were associated with an increased risk of coronary events including myocardial infarction and ischaemic stroke; the highest point estimate was noted for night shifts (RR = 1.41, 95% CI, 1.13–1.76) [10]. SW, specifically night and rotating shifts, has been identified as a risk factor for peptic ulcer when compared with regular day workers [13]. A Danish prospective study that followed a cohort of 20,000 nurses for 15 years concluded that night shift workers were at increased risk of developing type-2 diabetes when compared with their day shift counterparts [14]. A previous study investigated female shift workers and have suggested a connection between rotating SW and a delayed menopause [9]. Moreover, the International Agency for Research on Cancer (IARC) has classified night SW as a probable human carcinogen (Group 2A) [18, 19] with evidence of an association with prostate (RR = 1.24, 95% CI, 1.05–1.46) [15], colorectal (OR = 1.32, 95% CI, 1.12–1.55) [16], and breast (HR = 2.15, 95% CI, 1.23–3.73) [17] cancers.

Taken together, these studies support the concept that SW impacts worker's health significantly. While little is known about the physiological pathways underlying SW-related disease processes, several mechanisms have been hypothesized including circadian misalignment due to disturbed sleep, and light induced suppression of melatonin levels at night [20, 21]. These factors, in turn, disrupt a number of physiological and behavioral processes that contribute to disease progression. Any interference in regular circadian rhythm, due to sleep restriction, could result in disturbed metabolic, hormonal and inflammatory responses [22]. Such misalignment has been found to disturb cortisol levels, and pro- and anti-inflammatory proteins

[23], contributing to the development of chronic diseases including cognitive impairment. Another mechanism is endogenous melatonin rhythm. Melatonin is a hormone secreted from the Pineal body, low during the daylight hours and at the highest levels during dark periods (night) [20, 22, 24]. Exposure to light at night can reduce circulating melatonin levels [25]. If the light is bright, the levels can be completely suppressed, which may be a potential risk factor for chronic illnesses [26]. Shift workers experience substantial misalignment between circadian system of the body and unusual working schedules [27], which exposes them to increased risk for health problems. One more important factor that contributes to circadian interruption among shift workers is their behavior/lifestyle. This includes eating at irregular timings, lower physical activity levels, and higher incidence of smoking and intake of alcohol [28]. These factors feed back into the circadian clock causing desynchronized rhythms and altered metabolic and body temperature cycles [20, 27, 29]. In short, the potential effects of SW on cognition are likely related to the circadian misalignment and melatonin suppression. However, it is possible that any or all of the mechanisms described above interact to influence shift worker's cognitive function [27, 28].

The existing body of literature supports the notion that SW plays a critical role in cognitive function impairments. For example, some studies have highlighted acute and short term negative effects of night SW and disturbed sleep on cognitive functions [30–32]. In addition, few studies have assessed consequences of repeated disruption of circadian rhythms on cognitive functioning, among shift workers over time. First population based study [33] that explored chronic consequences of SW on cognitive functioning utilized large cross-sectional sample (3,237 French workers of aged 32, 42, 52, and 62 years) from VISAT (Aging, Health and Work) cohort. SW were classified into 'current', 'former' and 'never' and cognitive performances were evaluated by measuring verbal memory and speed performances. This first work provides some evidence of adverse effects of SW exposure on cognitive functioning, especially for men who are current or past shift workers [33]. Later, a prospective cohort study utilized the same French VISAT data base and reported that rotating SW was linked to lower cognitive test scores (memory and speed performances) [34]. According to the study, former shift workers who stopped working shifts within the previous five years showed improved cognitive functioning [34]. Furthermore, a cross-sectional Swedish EpiHealth cohort study (7,143 participants, aged between 45–75 years) investigated associations between history of SW (non-shift, past and current shift worker) and cognitive executive functions i.e., trail making test (TMT). The study revealed that current and former shift workers performed worse on TMT outcomes than non-shift workers [35]. More recently, analysis of 21,610 participants of aged 45–85 years, from the Canadian Longitudinal Study on Aging (CLSA) examined the cross-sectional relationship between SW (yes/no) and aspects of cognitive performance (declarative memory and executive functioning) [36]. The study found that shift workers showed poorer cognitive scores on tests for executive functioning (mental alteration test and interference condition of stroop test) but not for declarative memory (immediate and delayed recall trial) compared to non-shift workers [36]. Contrary to the results previously described, there are studies [37, 38] that did not find significant associations between SW and cognitive function. A sample of 595 participants with no dementia from a Swedish Adoption Twin Study of Aging (SATSA), were followed for 9 years [37]. No significant associations were reported between any types of SW (ever/never) and measures of cognitive performance (verbal, spatial, memory, processing speed, and general cognitive ability) [37]. Similarly, The Nurse's Health Study [38] followed 16,190 female participants in the United States, aged 30–55 years, over a 6-year period with three repeated cognitive assessments. Overall, this study does not convincingly support the hypothesis that older persons' cognition is negatively affected by their midlife SW history [38]. These discrepancies could be attributed to lack of differentiation between SW schedules

(night, rotating) [37], and restriction of study sample to highly educated participants, who held at least a registered nurse or bachelor's degree [38].

Despite SW's significance and crucial part in cognitive impairment, there is still a dearth of evidence. Existing normative standards utilized to evaluate cognitive functions were based on non-Canadian samples [33–35, 37, 39, 40], which may be outdated [33, 39, 40], and did not cover the full spectrum of ages from middle-aged to older adults [33, 34, 38–40]. Most previous studies [35–37] did not account for differentiation between SW schedules (night and rotating SW separately) [33, 35–38], and educational differences [34] in their analyses. Considering that circadian rhythm regulates cognitive activities [41, 42], desynchronization of the circadian rhythms associated with SW might be one of the plausible mechanisms underlying this association. Given the mixed findings reported from the limited studies on SW in relation to cognitive performance, this study aims to examine the association between SW and cognitive impairment measures based on normative standards from the Canadian population.

## Materials and methods

### Study design and sample

Cross-sectional data analyses were performed using the Canadian Longitudinal Study on Aging (CLSA) database, which is a nationwide, epidemiological study of aging that includes 51,338 middle-aged and older adults (aged 45–85 years). Participants were recruited and baseline data was collected between 2010 and 2015. The CLSA has two cohorts: the "tracking cohort" (21,241) that provided all information through telephone interviews, and a "comprehensive cohort" (30,097) that provided self-reported health information through in-home personal interviews, also, during a site visit at their local data collection center, they provided data including the entire neuropsychological battery. For this study, we combined both cohorts to increase sample size. CLSA excluded residents of institutions, the three territories, First Nations reserves; those who spoke neither English nor French; fulltime Canadian Armed Forces members; and individuals with overt cognitive impairment. The CLSA study design has been previously described in detail elsewhere [43, 44]. The core CLSA study has been approved by McMaster University Health Integrated Research Ethics Board and by research ethics boards at all collaborating Canadian institutions. The present study is a secondary analysis of fully de-identified CLSA data which has been approved by the York University, Office of Research Ethics (ORE) [STU 2020–123]. As such, additional participant consent for this analysis was not required as all CLSA participants provided informed consent during primary data collection to have their de-identified data used in research.

This study sample was limited to those who self-reported being currently employed or having previously worked. Fig 1 displays a flow diagram outlining the exclusion criteria. Participants were excluded if they reported: never working in any job; working in unspecified schedules; and refused to answer or do not know their working schedules. Finally, 47,811 participants remained in the study sample, which formed the basis of analysis.

### Assessment of primary exposure (SW)

The primary exposure of interest 'SW' was self-reported and assessed in the CLSA baseline questionnaires. All study participants were asked *"Have you ever worked at a job or business?"* (yes/no). The participants who reported 'yes' were asked *"Are you currently working at a job or business?"* (yes/no). Participants who reported 'yes' were asked *"Which of the following best describes your working schedule?"* (daytime work, night shift, rotating shift). Participants were also asked about their longest job, *"Thinking about the job you worked at the longest, which of*

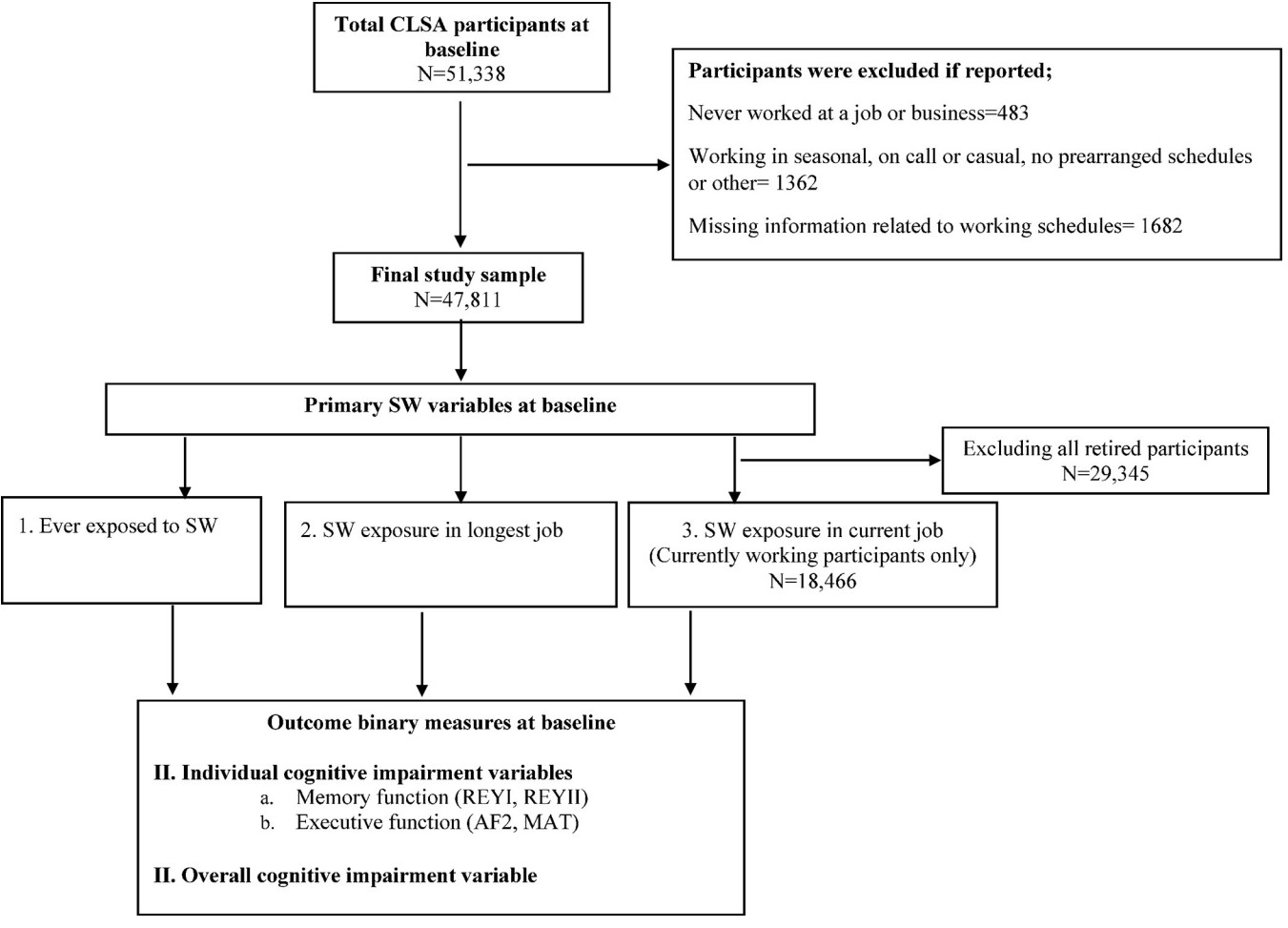

**Fig 1. Canadian Longitudinal Study on Aging participant flowchart.**

*the following best describes your working schedule*?" (daytime work, night shift, rotating shift). Based on this information, three variables were generated to measure SW exposure [9]:

**Ever exposed to SW.** This variable measured overall occurrence of any SW. All participants who reported ever worked in any shift (night/rotating) in their working career were considered exposed to SW and coded as 'yes'.

All those who reported only daytime work, were considered unexposed to SW and coded as 'no'.

**Exposure of SW in longest job.** This variable measured the exposure of SW during the longest job. SW in longest job was categorized into; daytime work (reference category), night SW and rotating SW.

**Exposure of SW in current job.** This variable measured the exposure of SW among participants who reported currently working and categorized into; daytime work (reference category), night SW, rotating SW.

## Primary outcome: Cognitive impairment

The primary outcome for this study is 'cognitive impairment', based on four cognitive function tests, including Rey Auditory Verbal Learning Tests (REYI and REYII) representing

**Table 1. Description of cognitive function tests utilized by CLSA to create cognitive impairment variables.**

| Cognitive domain assessed | Cognitive function tests | Cognitive function tests description | Individual impairment variables | Overall impairment variable |
|---|---|---|---|---|
| **Memory**<br><br>This domain assesses both learning and retention | Rey Auditory Verbal Learning Test (Immediate) (REYI)<br><br>Rey Auditory Verbal Learning Test (Delayed) (REYII) | The REYI is a 15-item word learning exam that measures learning as well as retention. This test is divided into two trials. During the first trial, 15 words were read out to the participants, who were then asked to immediately recall as many of the 15 words as they could in 90 seconds.<br><br>During second trial (REYII) participants were asked to recall as many of the 15 words from the first trial within 60 seconds after a 30 minutes delay. REY I and REY II were treated as separate memory tests. One point was allocated for each correctly recalled word in each of the tests. | Yes = Impaired on REYI if score is below the 5$^{th}$ percentile of healthy participants (normative data)<br><br>Yes = Impaired on REYII if score is below the 5$^{th}$ percentile of healthy participants (normative data) | Based on four individual impairment variables i.e. REYI, REYII, AF2, & MAT and can only be calculated when all scores are available.<br><br><br>Yes = Presence of two or more individual impaired scores are suggestive of overall impairment |
| **Executive Function**<br><br>This domain assesses many higher order mental processes and complex behaviours, including mental flexibility, abstract concept formation, problem-solving, and reasoning | Animal Fluency (AF2)<br><br>Mental Alteration (MAT) | Scores on the Animal Naming Test were based on a 'lenient' scoring algorithm where participants received one point for each distinct animal named, even if the animals were members of the same species.<br><br>Mental flexibility and processing speed are measured using the MAT. MAT involves a cognitive switching task that requires a participant to alternate between the numbers 1–26 and the letters of the alphabet (i.e. 1-A, 2-B, 3-C, etc.). | Yes = Impaired on AF2 if score is below the 5$^{th}$ percentile of healthy participants (normative data)<br><br>Yes = Impaired on MAT if score is below the 5$^{th}$ percentile of healthy participants (normative data) | |

After comparisons, the following impairment variables were created and utilized as outcome variables in the study

memory domain, with Animal Fluency (AF2), and Mental Alteration (MAT), representing the executive function domain of cognition. Details are summarized in Table 1. These domains were selected because they are present in both CLSA cohorts and each has been shown to correlate with everyday functioning (i.e. physical, behavioral, and social) [45–47]. The CLSA working group created normalized scores from the original test scores as standardized z-scores that have a mean of zero and standard deviation of 1.0. All normed z-scores were adjusted for age, sex and education status, and norming is done separately for tests completed in English and French. In order to determine whether a person's performance is within the range of healthy cognitive performance, comparisons were made with normative data. Normative data were created by CLSA [48] and used as a comparison standard for an individual's performance.

**Individual cognitive impairment variables.** Four binary-valued impairment variables (yes/no) were created, one for each of the cognitive function test i.e. REYI, REYII, AF2, and MAT. 'Yes' indicates that the participant's normed z-score falls in the lowest 5% of the neuro-healthy CLSA normative data.

**Overall cognitive impairment variable.** A binary-valued variable (yes/no) that indicates the participant's overall cognitive performance on the basis of four individual cognitive impairment variables and can only be calculated when all four are available i.e. REYI, REYII, AF2, and MAT [48, 49]. The presence of two or more individual impaired scores are suggestive of overall cognitive impairment and coded as 'yes', whereas zero or one impaired score is suggestive of 'no' impairment overall [48, 49].

Detailed rationale for the selection of these measures of 'cognitive impairment' along with explanation of their implementation and validity of utilized base rate algorithms have been published elsewhere [48, 50–54].

## Potential predictors

Multivariable models were adjusted for a relevant set of confounders. Potential confounders were determined by consulting the existing literature on SW and cognitive impairment [35–38]. Sociodemographic variables included, sex, age in years, ethnicity, education level, marital status, total annual household income in Canadian dollars, retirement status [35–38]. Lifestyle factors comprised of smoking status and alcohol consumption. Height and weight were used to calculate body mass index (BMI) in $kg/m^2$ [35–38]. Depression was ascertained by utilizing The Center for Epidemiological Studies short Depression scale (CES-D10). A score $\geq 10$ suggests the presence of depression [55, 56]. The measure of multi-morbidity was based on the standard definition [57] and included following chronic conditions: anxiety or mood disorder, Alzheimer's disease, arthritis, asthma, cancer, chronic obstructive pulmonary disease, diabetes, cardiovascular disease, and stroke. These chronic conditions were measured in the CLSA using the self-reported question, "Has a doctor ever told you that you have. . .?" The presence of $\geq 2$ chronic diseases were suggestive of multi-morbidity. CLSA measured Social Support Availability (SSA) by asking participants to rate their level of perceived support [58]. The measure contains 19 items rated on a 5-point Likert scale, from 1 (none of the time) to 5 (all of the time), with higher responses indicating better perceived support. Finally, the average SSA scores were used to create three categories, low, medium, and high [5]. 'Type of study cohort' is included as a covariate to represent the cohort membership, categorized into 'tracking' and 'comprehensive' [59]. Data from participants in the tracking cohort were collected over the phone, whereas data from participants in the comprehensive cohort were collected in-person [51]. All covariates were measured at baseline and their respective categories are summarized in Table 2.

## Analysis

Baseline characteristics were presented as frequencies. Unadjusted and adjusted logistic regression models were used to determine associations between SW variables and cognitive impairment individually, and also for overall cognitive impairment. Separate models were generated for each outcome variable. Odds ratios (ORs) with 95% confidence interval (CI) were calculated for all models. The CLSA provides survey inflation weights (i.e., inverse probability weights) and analytical weights, which were used for prevalence estimates and regression modeling respectively to generalize results to the Canadian population [60]. A P value less than 0.05 was considered statistically significant. To assess the robustness of our findings, sensitivity analyses were performed that compared complete cases included in analysis to the excluded cases due to missing information related to SW schedules, as well as cognitive impairment. Results from both sensitivity analyses are available in S1 Appendix. All statistical data analyses were performed using STATA version 13.0.

## Results

Fig 1 depicts a flow diagram summarizing the criteria used for exclusions. We compared the baseline characteristics of participants excluded due to missing information related to SW schedules (N = 1,682) with the participants included in our final study population (N = 47,811). No significant differences related to cognitive impairment variables were found between two groups. However, the participants excluded tend to be male, older (65 plus), less

**Table 2. Baseline characteristics of study sample and proportion of cognitive impairment indicators.**

| Characteristics | Total sample | Memory Function impairment | | Executive Function impairment | | Overall Impairment (%) [c,d] |
|---|---|---|---|---|---|---|
| | N[a] (%)[b] | REYI (%)[c,d] | REYII (%)[c,d] | AF2 (%) [c,d] | MAT (%) [c,d] | |
| Total study participants | 12,632,907 (100) | 6.44 | 5.73 | 6.11 | 9.10 | 4.40 |
| **Age (years)** | | | | | | |
| 45–54 | 4,887,556 (38.69) | 7.29 | 6.7 | 7.99 | 9.74 | 5.52 |
| 55–64 | 4,000,705 (31.67) | 5.68 | 5.28 | 5.35 | 8.00 | 3.98 |
| 65–74 | 2,367,052 (18.74) | 5.28 | 5.79 | 4.60 | 8.79 | 3.56 |
| 75+ | 1,375,571 (10.89) | 7.56 | 3.43 | 4.30 | 10.52 | 2.82 |
| **Sex** | | | | | | |
| Male | 6,141,058 (48.61) | 6.00 | 5.34 | 6.08 | 8.17 | 4.05 |
| Female | 6,491,848 (51.38) | 6.86 | 6.10 | 6.13 | 9.98 | 4.73 |
| **Ethnicity** | | | | | | |
| Whites | 11,989,923 (94.99) | 6.12 | 5.44 | 5.49 | 8.58 | 3.86 |
| Other[e] | 631,617 (5.01) | 12.49 | 11.13 | 17.71 | 19.91 | 14.46 |
| **Marital status** | | | | | | |
| With partner | 9,614,721 (76.13) | 5.96 | 5.51 | 5.72 | 8.39 | 3.99 |
| No partner | 3,014,661 (23.87) | 7.99 | 6.45 | 7.38 | 11.42 | 5.74 |
| **Education level** | | | | | | |
| Less than high school | 2,651,620 (20.99) | 7.31 | 4.69 | 5.26 | 10.80 | 3.55 |
| High school to some college | 7,519,899 (59.55) | 6.86 | 6.76 | 6.96 | 9.71 | 5.32 |
| Bachelor's degree and Higher | 2,456,491 (19.45) | 4.25 | 3.76 | 4.45 | 5.52 | 2.56 |
| **Household income (CAD)** | | | | | | |
| Less than $20,000 | 761,294 (6.36) | 13.57 | 11.39 | 9.70 | 16.99 | 12.4 |
| $20,000 or more, but less than $50,000 | 3,103,306 (25.94) | 6.5 | 5.54 | 7.10 | 11.91 | 5.04 |
| $50,000 or more, but less than $100,000 | 4,333,590 (36.22) | 5.99 | 5.41 | 5.75 | 8.37 | 3.94 |
| $100,000 and more | 3,765,059 (31.47) | 4.97 | 4.97 | 4.91 | 5.75 | 2.60 |
| **Smoking** | | | | | | |
| Never | 3,560,312 (28.25) | 5.99 | 5.23 | 5.85 | 8.73 | 4.65 |
| Former | 7,357,349 (58.39) | 6.19 | 5.54 | 5.99 | 8.53 | 3.77 |
| Current | 1,681,606 (13.34) | 8.41 | 7.60 | 7.24 | 12.56 | 6.52 |
| **Alcohol consumption** | | | | | | |
| Never | 1,945,792 (15.41) | 8.96 | 7.12 | 7.79 | 12.27 | 6.88 |
| Drinks less than weekly | 4,044,907 (32.05) | 7.18 | 6.05 | 6.58 | 9.83 | 5.13 |
| Drinks at least weekly | 6,630,183 (52.53) | 5.27 | 5.15 | 5.32 | 7.72 | 3.24 |
| **BMI (kg/m²)** | | | | | | |
| 20.0–24.99 (normal weight) | 3,834,669 (30.73) | 5.78 | 5.18 | 6.41 | 7.40 | 3.72 |
| <20.00 (underweight) | 419,736 (3.36) | 10.95 | 7.59 | 7.74 | 12.65 | 9.1 |
| 25.0–29.99 (overweight) | 4,800,165 (38.47) | 6.75 | 5.57 | 6.20 | 8.65 | 4.23 |
| >30.0 (obese) | 3,423,626 (27.43) | 6.17 | 6.39 | 5.34 | 11.04 | 4.67 |
| **Depression** | | | | | | |
| No (CES-D10 <10) | 10,330,178 (81.77) | 5.66 | 5.17 | 5.75 | 8.14 | 3.67 |
| Yes (CES-D10 ≥ 10) | 2,302,729 (18.22) | 9.97 | 8.32 | 7.72 | 13.45 | 7.68 |
| **Multi-morbidity** | | | | | | |
| Yes (≥2 chronic diseases) | 3,507,516 (27.76) | 7.67 | 6.72 | 5.62 | 10.89 | 5.29 |
| No (<2 chronic disease) | 9,125,391 (72.23) | 5.97 | 5.36 | 6.30 | 8.43 | 4.06 |
| **Social Support Availability (SSA)** | | | | | | |
| Low | 4,009,023 (32.97) | 8.55 | 6.68 | 7.30 | 10.93 | 5.99 |

*(Continued)*

**Table 2.** (Continued)

| Characteristics | Total sample | Memory Function impairment | | Executive Function impairment | | Overall Impairment (%) [c,d] |
|---|---|---|---|---|---|---|
| | N[a] (%)[b] | REYI (%)[c,d] | REYII (%)[c,d] | AF2 (%) [c,d] | MAT (%) [c,d] | |
| Medium | 3,878,890 (31.90) | 5.62 | 4.75 | 5.73 | 8.43 | 3.57 |
| High | 4,271,457 (35.13) | 5.07 | 5.50 | 5.28 | 7.74 | 3.55 |
| **Retirement status** | | | | | | |
| Completely /partially retired | 5,951,165 (47.34) | 6.31 | 5.54 | 4.85 | 9.44 | 4.13 |
| Not retired | 6,618,004 (52.65) | 6.53 | 5.90 | 7.23 | 8.84 | 4.61 |
| **Type of study cohort** | | | | | | |
| Tracking | 5,156,753 (40.82) | 6.23 | 5.56 | 5.95 | 9.14 | 4.28 |
| Comprehensive | 7,476,154 (59.18) | 7.07 | 6.26 | 6.57 | 8.98 | 4.72 |
| **Primary SW exposure variables** | | | | | | |
| **Ever exposed to SW** | | | | | | |
| Never exposed to SW(Daytime work only) | 9,631,177 (78.81) | 6.12 | 5.43 | 5.89 | 8.29 | 4.04 |
| Ever exposed to SW | 2,589,578 (21.19) | 7.89 | 7.24 | 7.31 | 12.21 | 6.10 |
| **SW exposure in longest job** | | | | | | |
| Not exposed to SW (Daytime work) | 9,620,140 (80.54) | 6.11 | 5.43 | 5.89 | 8.28 | 4.04 |
| Night SW | 460,950 (3.86) | 9.74 | 9.03 | 7.69 | 13.22 | 8.68 |
| Rotating SW | 1,863,147 (15.60) | 7.49 | 6.63 | 7.23 | 12.20 | 5.29 |
| **SW exposure in current job[f]** | | | | | | |
| Not exposed to SW (Daytime work) | 4,905,640 (84.38) | 5.72 | 5.36 | 7.00 | 7.18 | 3.90 |
| Night SW | 235,468 (4.05) | 11.21 | 5.16 | 9.78 | 11.53 | 7.62 |
| Rotating SW | 672,753 (11.57) | 8.3 | 6.43 | 7.37 | 12.02 | 5.26 |

[a] Reported sample size is an estimation for the study sample (N = 47,811) using survey inflation weights; figures do not always sum up to the total due to missing values

[b] Reported frequencies are column percentages, calculated using survey inflation weights

[c] Reported frequencies are row percentages, calculated using survey inflation weights

[d] Prevalence of cognitive impairment was based on normalized scores with respect to participant's age, sex, education and within each language group (English and French).

[e] Other included South Asian, Chinese, Filipino, Latin American, Japanese, Southeast Asian, Korean, Arab, West Asian, and Black.

[f] For current job, only those participants were included who reported currently working (not retired) at baseline (N = 18,466)

CAD, Canadian dollars; BMI, body mass index; CES-D, the Center for Epidemiological Studies Depression Scale SW, shift work; REY, Rey auditory verbal learning; AF, Animal fluency; MAT, Mental alteration.

educated, non consumer of alcohol, and belongs to comprehensive cohort. Details are available in S1 Appendix. We also compared the participants with missing information related to cognitive impairment (N = 1,201) to the participants with complete cognitive impairment data (N = 46,610). No significant differences were found related to SW exposure between two groups. However, the participants with missing information tend to be in older group (65 plus), retired and living without partner, belong to tracking cohort and low income group, consume less alcohol, have depression and multi morbidity. Details are available in S1 Appendix.

Table 2 shows summary statistics of 47,811 participants (weighted to represent 12,632,907 Canadians) at baseline. The mean age of participants was 59.7 years (SD, 10.15 years), and 51.4% were females. Most participants (95%) were White, more than 50% reported to be living with partners, having education of high school to some college level, former smokers, drinking at least weekly, still working (not retired) and had household income 50,000 CAD and more. Around 30% were obese, had multiple-morbidity, and reported low to medium SSA. Overall,

**Table 3. Unadjusted logistic regression [odds ratios (ORs) and 95% Confidence Intervals(CI)] for individual and overall cognitive impairment for primary SW variables.**

| Primary SW variables | Memory | | Executive function | | Overall |
|---|---|---|---|---|---|
| | REYI | REYII | AF2 | MAT | |
| | Impaired cognition OR (95% CI) [a] | Impaired cognition OR (95% CI) [a] | Impaired cognition OR (95% CI) [a] | Impaired cognition OR (95% CI) [a] | Impaired cognition OR (95% CI) [a] |
| **Ever exposed to SW** | | | | | |
| Never exposed to SW (Daytime work only) | 1.00 | 1.00 | 1.00 | 1.00 | 1.00 |
| Ever exposed to SW | 1.12 (0.98–1.29) | **1.23 (1.06–1.41)*** | **1.36 (1.18–1.56)*** | **1.41 (1.25–1.59)*** | **1.42 (1.2–1.69)*** |
| **SW exposure in longest job** | | | | | |
| Not exposed to SW (Daytime work) | 1.00 | 1.00 | 1.00 | 1.00 | 1.00 |
| Night SW | 1.29 (0.96–1.73) | **1.79 (1.34–2.39)*** | **1.74 (1.25–2.41)*** | **1.78 (1.37–2.29)*** | **2.26 (1.58–3.23)*** |
| Rotating SW | 1.09 (0.93–1.28) | 1.11 (0.94–1.29) | **1.24 (1.07–1.45)*** | **1.35 (1.18–1.54)*** | **1.24 (1.02–1.49)*** |
| **SW exposure in current job[b]** | | | | | |
| Not exposed to SW (Daytime work) | 1.00 | 1.00 | 1.00 | 1.00 | 1.00 |
| Night SW | **1.81 (1.25–2.62)*** | **1.53 (1.01–2.31)*** | **1.88 (1.26–2.79)*** | **1.99 (1.36–2.93)*** | **2.85 (1.74–4.67)*** |
| Rotating SW | 1.18 (0.90–1.55) | 1.16 (0.88–1.51) | 1.11(0.87–1.41) | **1.48(1.17–1.86)*** | 1.26 (0.92–1.73) |

* *P* value <0.05

[a] The ORs and 95% CI were calculated using survey analytical weights.

[b] For current job, only those participants were included who reported currently working (not retired) (N = 18,466)

one in every five individuals (21.1%) reported having been exposed to some kind of SW at work. 4% and 11.6% of currently working participants, respectively, reported being exposed to night and rotating SW. Considering the longest job held in their whole career, 3.9% and 15.6% of individuals reported being exposed to night and rotating SW, respectively.

The proportions for cognitive impairment (both individual and overall) are included in Table 2. Higher cognitive impairment was noted among those who reported ever exposed to any type of SW compared to those never exposed (daytime work only). Consistently, individuals who reported being exposed to night and rotating SW during current or longest job had a greater proportion of cognitive impairment compared to those who only reported daytime work. Unadjusted logistic regression analyses indicated that SW exposures were associated with higher odds of cognitive impairment compared to those who were unexposed (Table 3). Results of multivariate analyses suggested that SW exposure was related to increased odds of cognitive impairment, after adjustments for confounders (Table 4).

## Relationship between SW exposure and overall cognition impairment

Separate models were constructed to evaluate the associations between SW exposures and overall cognitive impairment. For ever exposed to SW, no significant associations was observed for overall impairment (OR, 1.12; 95% CI, 0.92–1.35). However, overall cognitive impairment was found significant among participants who reported to be exposed to night SW during their current job (OR, 1.79; 95% CI, 1.08–2.96) and, night SW during their longest job (OR, 1.53; 95% CI, 1.04–2.26) when compared to those who only reported day time work (Table 4).

Results of the multivariate analysis also exposed significant associations among employed participants between sociodemographic factors and overall cognitive impairment (estimators for confounders are not shown in Table 4, details are available in S2 Appendix). In general,

**Table 4. Adjusted logistic regression [odds ratios (ORs) and 95% Confidence Intervals(CI)] for individual and overall cognitive impairment for primary SW variables.**

| Primary SW variables | Memory function | | Executive function | | Overall |
|---|---|---|---|---|---|
| | REYI | REYII | AF2 | MAT | |
| | Impaired cognition OR (95% CI) [a,b] | Impaired cognition OR (95% CI) [a,b] | Impaired cognition OR (95% CI) [a,b] | Impaired cognition OR (95% CI) [a,b] | Impaired cognition OR (95% CI) [a,b] |
| **Ever exposed to SW** | | | | | |
| Never exposed to SW (Daytime work only) | 1.00 | 1.00 | 1.00 | 1.00 | 1.00 |
| Ever exposed to SW | 0.95 (0.81–1.11) | 1.08 (0.92–1.26) | 1.12 (0.96–1.30) | **1.14 (1.00–1.29)**\* | 1.12 (0.92–1.35) |
| **SW exposure in longest job** | | | | | |
| Not exposed to SW (Daytime work) | 1.00 | 1.00 | 1.00 | 1.00 | 1.00 |
| Night SW | 0.95 (0.67–1.33) | **1.44 (1.03–2.01)**\* | 1.27 (0.92–1.76) | 1.20 (0.90–1.60) | **1.53 (1.04–2.26)**\* |
| Rotating SW | 0.97 (0.82–1.16) | 0.99 (0.82–1.16) | 1.07 (0.90–1.26) | **1.16 (1.01–1.34)**\* | 1.02 (0.83–1.27) |
| **SW exposure in current job[c]** | | | | | |
| Not exposed to SW (Daytime work) | 1.00 | 1.00 | 1.00 | 1.00 | 1.00 |
| Night SW | 1.47 (0.97–2.21) | 1.28 (0.81–2.05) | 1.32 (0.87–1.99) | 1.31 (0.87–1.99) | **1.79 (1.08–2.96)**\* |
| Rotating SW | 1.06 (0.79–1.41) | 1.03 (0.77–1.36) | 0.95 (0.73–1.26) | **1.36 (1.06–1.74)**\* | 1.04 (0.73–1.17) |

\* *P* value <0.05

[a] The ORs and 95% CI were calculated using survey analytical weights.

[b] Models are adjusted for age, sex, ethnicity, marital status, education, income, BMI, smoking, alcohol consumption, retirement status, depression, multi-morbidity, social support availability index, type of study cohort. Estimators for all confounders are not presented here, and are available in S2 Appendix

[c] For current job, only those participants were included who reported currently working (not retired) (N = 18,466),and the models are adjusted for all covariates mentioned above except for retirement status

SW, shift work; OR, odds ratio; CI, confidence interval.

overall cognitive impairment was significantly higher among non-white workers (OR, 4.83; 95% CI, 3.55–6.57), workers having depression (OR, 1.80; 95% CI, 1.38–2.34) and having education of high school to some college (OR, 2.37; 95% CI, 1.69–0.83). However, lower odds of overall cognitive impairment were noted for those workers who belong to older age groups (55 and plus) (OR, 0.65; 95% CI, 0.51–0.83), higher income groups (50,000 CAD and above) (OR, 0.48; 95% CI, 0.37–0.63), having some social support (OR, 0.81; 95% CI, 0.67–0.97), and drinks at least weekly (OR, 0.74; 95% CI, 0.59–0.91). Details of complete models including estimators for all confounders are presented in separate tables (S2 Appendix).

### Relationship between SW exposure and memory function of cognition

The associations between SW exposures and memory function of cognitive impairment (REYI and REYII) were examined and separate models were constructed. For REYI measure of memory function, no statistically significant results were found across all primary SW variables (Table 4). Based on REYII measure of memory function (Table 4), participants who were exposed to night SW during their longest job were found significantly associated with cognitive impairment compared to those reported only daytime work (OR, 1.44; 95% CI, 1.03–2.01). The association remained non-significant for current night (OR, 1.28; 95% CI, 0.81–2.05) and rotating (OR, 1.03; 95% CI, 0.77–1.36) shift workers.

In addition, the multivariate analysis revealed substantial associations between sociodemographic characteristics and memory function impairment among employed participants

(estimators for confounders are not shown in Table 4, details are available in S2 Appendix). Memory function impairment (both REYI and REYII) was significantly higher among non-white workers (OR, 2.12; 95% CI, 1.57–2.86), and workers having depression (OR, 1.31; 95% CI, 1.13–1.52). However, lower odds of memory function impairment (both REYI and REYII) were noted for those workers who belong to older age groups (OR, 0.46; 95% CI, 0.35–0.60), and higher income groups (20,000 CAD and above) (OR, 0.57; 95% CI, 0.44–0.73). REYI impairment reduced for those having social support (OR, 0.81; 95% CI, 0.70–0.94), having high education (OR, 0.74; 95% CI, 0.58–0.93) and those who drink at least weekly (OR, 0.75; 95% CI, 0.63–0.89). Details of complete models including estimators for all confounders are presented in separate tables (S2 Appendix).

### Relationship between SW exposure and executive function of cognition

Separate models were generated to evaluate the relationships between SW exposures and executive function impairment (AF2 and MAT). For AF2 measure of executive function, no statistically significant results were noticed across all primary SW variables (Table 4). However, based on MAT measure of executive function (Table 4), cognitive impairment was associated with participants who reported ever exposed to any type of SW (OR, 1.14; 95% CI, 1.00–1.30), exposed to rotating SW in their current job (OR, 1.36; 95% CI, 1.06–1.74), and exposed to rotating SW in their longest job (OR, 1.16; 95% CI, 1.01–1.34) compared to those who had never been exposed to SW (daytime work only). For the AF2 measure of executive function, no statistically significant results were found across all primary SW variables (Table 4).

Furthermore, among participants who were employed, the multivariate analysis showed significant associations between sociodemographic factors and executive function impairment (estimators for confounders are not shown in Table 4, details are available in S2 Appendix). Executive function impairment (both AF2 and MAT) was significantly higher among non-white workers (OR, 3.06; 95% CI, 2.39–3.93), and workers having depression (OR, 1.34; 95% CI, 1.10–1.50). However, lower odds of executive function impairment (both AF2 and MAT) were noted for those workers who belong to older age groups (OR, 0.76; 95% CI, 0.64–0.91), and higher income groups (20,000 CAD and above) (OR, 0.57; 95% CI, 0.34–0.98). Also, MAT based cognitive impairment was found significantly higher for workers who are current smokers (OR, 1.34; 95% CI, 1.10–1.50), and obese (OR, 1.35; 95% CI, 1.08–1.68). Details of complete models including estimators for all confounders are presented in separate tables (S2 Appendix).

### Discussion

The purpose of this study was to investigate the associations between SW exposure and cognitive impairment among middle-aged and older adults. The results of this study demonstrated that SW exposure has significant relationship with cognitive impairment. Overall cognitive impairment was evident for those exposed to night SW, both during current and longest job, compared to those who worked daytime shift only. In terms of domain-based measures, night SW exposure in longest job was related to memory function impairment and those exposed to rotating SW, both in current and longest job, were more likely than daytime workers to have impaired executive function. These findings are clinically relevant and support the notion that circadian misalignment would render shift workers more vulnerable to cognitive impairment. It is imperative to identify and comprehend modifying risk factors, like SW, associated with cognitive impairment, since this is critical for designing and implementing suitable prevention strategies.

Globally, SW is prevalent and these study results are consistent with the literature [7–9] indicating that 21% of Canadians were exposed to some kind of SW during their career.

Findings from other developed economies such as France [61], Japan [62], across Europe [63] and the United States [63], also confirms similar pattern where 20% to 25% of workers were exposed to SW in various sectors. Moreover, this study reported the overall proportion of cognitive impairment as 4.4%, which is consistent with rates observed (5.3% and 2.8%) in some previous studies [64, 65]. However, other studies [66–68], have discovered higher rates (10.8%, 7%, 8.7%) of cognitive impairment. A major factor that may have influenced these differences is the CLSA study design, which included only persons without overt cognitive impairment at baseline [49].

The study results suggest that the association between night SW (both in current and longest job) and overall cognitive impairment were significant. As far as domain-based measures are concerned, night SW (in longest job) and rotating SW (both in current and longest job) were more likely than daytime workers to have impaired memory (REYII) and executive function (MAT) respectively. These findings support evidence from previous research linking SW with cognitive impairment. A population based study [33] that explored chronic consequences of SW (current, former or never) reported that cognitive functions among former shift workers tends to be impaired. Later, a prospective cohort study also stated that rotating SW was associated with lower cognitive test scores [34]. Similarly, a cross-sectional Swedish study linked a history of any SW to lower cognitive performances and observed that current shift workers performed worse on the cognitive tests than non-shift workers [35]. Contrary to the results previously described [33–35] other studies [37, 38] did not find significant associations between SW and cognitive function. Possible explanations for these contrasting results might be due to differences in the classification of SW (24), [36] (unable to account for types of SW), age categorization [38] (did not cover full spectrum and limited to age group 58–68 years), and the use of a non-representative highly educated sample [38] (restricted to nurses).

Recently, Alonzo et al. [36] documented lower performance on measures of executive function (MAT) among shift workers. However, the measures based on declarative memory (REYI and REYII) did not find any statistically significant results, whereas our adjusted analysis indicated that exposure to night SW during longest job is associated with impaired measure of memory function (REYII). The results of Alonzo et al. [36] need to be interpreted with caution as several limitations have been identified. First, measures used to assess the cognitive function were based on scores without any demographic adjustments of age, sex, education and language. Exploration of cognition in an aging population without adjustment for demographic variables associated with healthy aging will produce misleading results due to measurement bias [48, 69]. In contrast, our study utilized 'normalized cognitive scores' with respect to participant's age, sex, education and within each language group (English and French), hence reducing the chance of measurement bias [48]. Second, findings lacked comparisons with normative data, which is fundamental for the interpretation of neuropsychological test scores and determining whether a person's performance is below the range of healthy cognitive performance. Finally, their study [36] did not examine types of SW separately, i.e., night and rotating SW. In contrast, our analysis takes into account different types of SW (night and rotating SW) for both current and longest job. The relationship between SW and cognitive functions may not be same across different types of SW (night and rotating SW). Also, rotating SW has been hypothesized to be more disruptive to circadian rhythm than regular night work and it is possible that rotating shift workers demonstrate greater difficulty in adapting to work schedules as they have to move from shift to shift compared with regular night shift workers [70].

Although the relationship between SW and cognitive performances is inconclusive [3] there are good reasons to believe that such a relationship may exist. One possible pathophysiological mechanism underlying the association between SW, and cognitive impairment has been thought to be the repeated desynchronization of body clock due to working and sleeping

at the wrong circadian phase among shift workers [34, 71, 72]. This could demonstrate harmful impacts on health, such as sleep deprivation, daytime sleepiness, and brain inflammation, making people more susceptible to cognitive decline [33, 38, 71, 73, 74]. Another mechanism is the repeated physiological stress and increased levels of cortisol induced by circadian disruption, as evidenced by a previous study [75] which explored the influence of chronic jet lag on cognitive functions among airline cabin crew. Moreover, disturbed circadian rhythms have also been linked to neurodegeneration [76, 77]. It is possible that impaired pineal secretion of melatonin, due to unusual light exposures among shift workers, may significantly impair the normal antioxidant defenses of the brain, contributing to cognitive impairment [78]. On balance, the literature supports the notion that circadian disruption due to SW plays a critical role in cognitive functions. However, additional studies are needed to confirm the association between SW and cognitive impairment, as well as any physiological pathways that underlie the mechanism.

Results of the multivariate analysis also revealed substantial associations between sociodemographic factors among shift workers and cognitive impairment (S2 Appendix). Consistent with literature [79, 80] non-white ethnicity in our study sample was significantly associated with cognitive impairment. Similarly, workers who were current smokers [81], have depression [82] and have higher BMI were associated with higher odds of cognitive impairment. An explanation for these findings is that the smoking can cause periventricular and subcortical white matter lesion progression [83], cholinergic system in the basal forebrain can be effected by depression [84], and obesity can cause local inflammation within the hypothalamus that alters synaptic plasticity, thus contributing to neurodegeneration [85]. In addition, high income groups and better social support were related to reduce cognitive impairment. Employment and income levels are indicators of economic security as well as social and psychological stress, which can affect brain function and cognition [5, 86, 87]. However, lower odds of cognitive impairment among older age groups in our study is contrary to what was previously reported [35, 37, 88]. This inconsistency may be due to exclusion of persons with overt cognitive impairment from CLSA database at baseline, as a result cognitively healthy subgroup of the population may have chosen to participate. This is probably why the overall proportion of participants with cognitive impairment decreases as age group increases.

There are some additional occupational characteristics (not included in this study) that have been linked to cognitive impairment. According to prior studies [89, 90], high mental demands at work are significantly associated with better cognitive functioning in old age. Despite the possibility that work-related stress brought on by complex work, such as inadequate job control, high job demands, a lack of social support, and manual labour, may raise the risk of dementia [91, 92], research revealed that high work complexity was associated with a lower risk of dementia [93, 94]. Cognitive reserve in workers with higher work complexity can serve as a neuroprotective agent thus postponing the cognitive decline [95]. There is an increased risk of cognitive impairment due to certain occupational and environmental exposures that are neurotoxic to brain cells, such as lead [96], organophosphate pesticides [97], and magnetic fields among electronic workers [98].

Our study had several strengths. A major strength of this study is that, to our knowledge, it is the first study to investigate the associations between different types of SW exposure (night and rotating shift), and cognitive impairment based on Canadian standards, which means that cognitive impairment was identified after comparisons with neuro-healthy normative data [48, 49]. All cognitive test scores are normalized for the participant's age, sex, education level and language of test administration (English and French) [48]. Such normalization of cognitive scores and comparisons with normative data were lacking in a previous study [36], and are required to determine whether a person's performance falls within the range of healthy

cognitive performance [48]. In addition, a large population based sample was used involving a wide range of participants. Nonetheless, there are limitations to this study that are worth noting. There were some differences in mode of data collection between tracking and comprehensive cohorts that is phone vs. in-person respectively. We controlled for type of study cohort in all adjusted models as this approach has been previously utilized [59], reducing the potential for this to have affected the study findings. Due to the number of events evening and night shifts were pooled together as previously done by some researchers [8, 11, 99]. Some SW related information were not included, as they were not recorded in the CLSA questionnaire, such as the type and direction of rotating shifts, number of consecutive night shifts worked, and the number of days off between shifts [26]. Type and duration of jobs were not examined and the association between SW and cognitive functions may not be constant across all types and duration of jobs [100, 101]. The lack of this information is a limitation and suggest potential areas for future investigation. Some participants were excluded from analysis due to missing information related to SW schedules (N = 1,682; 3.2%) and cognitive impairment (N = 1,201; 2.3%). Despite the relatively small proportion of missing data, there were some statistically significant differences between the missing and complete cases (S1 Appendix), which may result in potential bias. Respondents were free of overt cognitive impairment at baseline and are more likely healthier than the regular population possibly leading to an underestimation of the magnitude of some of our findings. In addition, generalizability of the results is limited to those healthier than the overall population. Moreover, due to the cross-sectional nature of our study, we are unable to assess temporality in the relationship between SW and cognitive impairment, raising the possibility of reverse causation.

## Conclusion

These findings highlight the negative impact of SW on cognitive function in middle-aged and older adults. By taking this modifiable risk factor into account we may enable workers to reduce cognitive impairment both during their working lives and after retirement, and support "active aging" of the workforce. Although these findings are preliminary, they suggest that SW exposure and circadian disruption may be an important factor in the risk of cognitive impairment and warrants further investigation.

## Supporting information

**S1 Checklist. STROBE statement—checklist of items that should be included in reports of observational studies.**
(DOCX)

**S1 Appendix. Comparisons of complete versus missing cases.**
(DOCX)

**S2 Appendix. Adjusted logistic regression models with estimates for all covariates.**
(DOCX)

## Acknowledgments

This research was made possible using the data/biospecimens collected by the Canadian Longitudinal Study on Aging (CLSA). This research has been conducted using the CLSA dataset Baseline Tracking Dataset version 4.0, Baseline Comprehensive Dataset version 7.0, under Application Number 2010006. The CLSA is led by Drs. Parminder Raina, Christina Wolfson and Susan Kirkland.

## Author Contributions

**Conceptualization:** Durdana Khan, Hala Tamim.

**Formal analysis:** Durdana Khan, Hala Tamim.

**Investigation:** Durdana Khan, Hala Tamim.

**Methodology:** Durdana Khan, Hala Tamim.

**Supervision:** Heather Edgell, Michael Rotondi, Hala Tamim.

**Writing – original draft:** Durdana Khan.

**Writing – review & editing:** Durdana Khan, Heather Edgell, Michael Rotondi, Hala Tamim.

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
