## [Decision Letter · Decision Letter 0]

15 Jun 2023

PONE-D-23-11981The association between shift work exposure and cognitive impairment among middle-aged and older adults: results from the Canadian Longitudinal Study on Aging (CLSA)PLOS ONE

Dear Dr. Khan,

Thank you for submitting your manuscript to PLOS ONE. After careful consideration, we feel that it has merit but does not fully meet PLOS ONE’s publication criteria as it currently stands. Therefore, we invite you to submit a revised version of the manuscript that addresses the points raised during the review process.

Both reviewers recommend a more comprehensive presentation of the theoretical basis and mechanisms linking shift work and cognitive impairment and of the negative health consequences associated with shift work. I share this view and find that the influence of circadian rhythm desynchronization on shift work, in particular, is too brief. In general, you give a lot of references in the introduction, but they are not very precise. Please revise this in the introduction as well. Also make sure that the bibliography is correct (e.g. Alonzo et al. (55) is not correct).

From my point of view, Table 4 represents the most important results. Therefore, in addition to the reviewer requirements, I would like you to present these results in more detail. Also, the estimators for the confounders in the multiple models should be presented. This will allow the reader to see whether shift work or confounders have a stronger impact on cognitive function.

We look forward to receiving your revised manuscript.

Kind regards,

Swaantje Wiarda Casjens

Academic Editor

PLOS ONE

Journal Requirements:

- 10.1097/GME.0000000000001981

In your revision ensure you cite all your sources (including your own works), and quote or rephrase any duplicated text outside the methods section. Further consideration is dependent on these concerns being addressed.

Reviewers' comments:

Reviewer's Responses to Questions

**Comments to the Author**

1. Is the manuscript technically sound, and do the data support the conclusions?

Reviewer #1: Yes

Reviewer #2: Yes

2. Has the statistical analysis been performed appropriately and rigorously? 

Reviewer #1: Yes

Reviewer #2: Yes

3. Have the authors made all data underlying the findings in their manuscript fully available?

Reviewer #1: Yes

Reviewer #2: No

4. Is the manuscript presented in an intelligible fashion and written in standard English?

Reviewer #1: Yes

Reviewer #2: Yes

5. Review Comments to the Author

Reviewer #1: This is a very interesting and succinct secondary analysis exploring the association between shift work and cognitive impairment using data from the CLSA. Please see the attached document with my comments.

Reviewer #2: Comments to the authors

The manuscript entitled “The association between shift work exposure and cognitive impairment among middle aged and older adults: results from the Canadian Longitudinal Study on Aging (CLSA)” investigated the possible correlation between shift work exposure and cognitive impairment in a large sample of Canadian population. This is a particularly interesting and also topical issue from the point of view of both occupational medicine and public health. Indeed, recently several studies suggested that sleep disturbance is predictive of cognitive decline in older people and such problems should be identified and treated early to prevent a deterioration of cognitive functions. On the other hand, shift workers, especially those who carry out night work, experience an average reduction of sleep of about two hour and working atypical shifts, outside the normal daylight hours (7/8 am–5/6 pm), which can encompass early-morning, afternoon, evening or night shifts might lead to the shift work sleep disorder that is a condition characterized by trouble sleeping, excessive sleepiness and fatigue. Therefore, although an increasing number of studies over the last few years have suggested that shift work (particularly that including working at night) and in some cases also long working hour are able to cause important modifications of cognitive functions, several issues, such as which cognitive domains are most affected and to what extent, the identification of pathophysiological mechanisms underlying the observed effects, the assessment of a potential dose-response relationship, the possibility of functional recovery after leaving shift work and the acknowledgement of the impact correlated to chronic exposure, still need to be clarified. In this regard, and in my opinion, the findings provided by this study are a useful contribution to deepening knowledge on this topic and improving management strategies for workers exposed to shift or night work. The study is generally well-designed but the authors should only address the followings minor comments and/or suggestions in order to improve the manuscript:

Abstract

The methods section generically refers to cognitive tests. I think it would be appropriate to make explicit which tests were used.

Introduction

I found the introduction very comprehensive and well structured. The authors describe the main characteristics of shift works, report literature data indicating potential adverse effects resulting from exposure to this occupational risk factor and their supposed mechanisms of action so that the reader can easily understand the rationale and aim of the study. Therefore I have only few and minor comments and suggestions:

• Pag. 3, lines 55-56: “A variety of negative health outcomes have been associated with SW, particularly night and rotating SW”. I think readers would be interested in a short, concise but comprehensive list of what adverse effects we are talking about;

• Pag 3, lines 56-57: “The existing body of literature supports the notion that SW plays a critical role in cognitive functions.”. In my opinion this sentence should be reformulated for clarity… maybe the authors meant “… in cognitive function impairment”;

Discussion

The authors, referring to the limits of their study, correctly stated that “…type and duration of job were not examined and the association between SW and cognitive functions may not be constant across all types and duration of job (66,67). Third, some SW related information were not included, as they were not recorded in the CLSA questionnaire, such as the type and direction of rotating shifts, number of consecutive night shifts worked, and the number of days off between shifts”.

I suggest to include in the “Discussion” section a brief paragraph in which the authors discuss how the aforementioned SW characteristics could affect the cognitive functions of exposed workers. In addition, to provide a more comprehensive overview, it might be useful to add a few sentences about what other occupational risk factors (besides shift work) might have adverse effects on the cognitive functions of exposed workers (psychosocial risk factors? heavy metals? solvents? other?)

Minor Revisions

Page 9, Table 2, last column heading: Please correct “impairment”;

Page 13, line 197: “…to be exposed to;”. Please delete the semicolon;

Page 13, line 207: “…who reported;”. Please delete the semicolon;

6. PLOS authors have the option to publish the peer review history of their article (what does this mean?). If published, this will include your full peer review and any attached files.

Reviewer #1: No

Reviewer #2: No

---

## [Author Response · Author response to Decision Letter 0]

7 Jul 2023

Swaantje Wiarda Casjens

Academic Editor

PLOS ONE

Thank you for the editor's and reviewers’ suggestions. We are confident that the changes have improved our manuscript. Please see the answers to the comments below. All changes have been incorporated in the revised manuscript. 

Academic editor’s comments and author responses

Comment 1: 

Both reviewers recommend a more comprehensive presentation of the theoretical basis and mechanisms linking shift work and cognitive impairment and of the negative health consequences associated with shift work. I share this view and find that the influence of circadian rhythm desynchronization on shift work, in particular, is too brief. In general, you give a lot of references in the introduction, but they are not very precise. Please revise this in the introduction as well. 

Authors: 

Thank you for raising a valid point. We have added more comprehensive details regarding underlying pathophysiological mechanisms linking shift work and negative health outcomes including cognitive impairment to the introduction and discussion sections of revised manuscript. In addition we have added details of all negative health consequences associated with shift work in the introduction section of the revised manuscript.

Comment 2:

Also make sure that the bibliography is correct (e.g. Alonzo et al. (55) is not correct).

Authors:

Thank you for pointing this out. We have corrected this error in the revised manuscript. In the revised manuscript the study ( Alonzo et al.) reference is #36. .

Comment 3:

From my point of view, Table 4 represents the most important results. Therefore, in addition to the reviewer requirements, I would like you to present these results in more detail. Also, the estimators for the confounders in the multiple models should be presented. This will allow the reader to see whether shift work or confounders have a stronger impact on cognitive function.

Authors:

Thank you. You have raised an important point. We agree that Table 4 represents important results and they should be discussed in detail and in relation to estimators for the confounders. We have presented further details in the result section of the revised manuscript. Also, we have elaborated and clearly explained the significant estimators of the confounders in each of the result section. We again highlighted the significant confounders and discuss their relationship with the cognitive impairment in discussion section of revised manuscript.

Furthermore, we have added five tables (four for individual cognitive impairment, one for overall cognitive impairment) which contain details of complete multivariate logistic regression models including estimator details of confounders in S2 Appendix, as your point is well taken. 

We agree that the details will allow the readers to see and understand which shift work or confounders have a stronger impact on cognitive function.

Reviewer comments and author responses

Reviewer #1

Comment 1:

The theoretical foundations and mechanisms linking shift work and cognitive impairment should be laid out and described in the introduction. I observe that the authors do mention these mechanisms in various places in the manuscript (e.g., desynchronization of the circadian rhythm on p. 3), though readers may find it helpful to have the theory laid out at the beginning of the manuscript. The authors may then circle back in the discussion to identify which theory or mechanism appears to be most supported by the results.

Authors:

Thank you. The reviewer raised an important point. We have added more comprehensive details regarding underlying pathophysiological mechanisms linking shift work and negative health outcomes, including cognitive impairment, to the introduction section of the revised manuscript. Also, we have elaborated on the mechanisms again in the discussion section of the revised manuscript with more focus on mechanisms linking shift work and cognitive impairment. 

Comment 2:

The literature review could benefit from more detail regarding the characteristics of the published studies. For example, what were the age ranges of participants in these studies, how was cognition measured, what were the sample sizes, were these studies cross-sectional or longitudinal, etc.? These additional details could help the authors identify reasons for discrepancies between studies and set the stage for them to more succinctly explain the novelty of their work and its contribution to the literature.

Authors:

Thank you. The reviewer makes an important point. We have added and clearly explained the relevant details (age groups, database utilized, sample sizes, study designs and details related to shift work and cognitive measures) of published studies in the introduction section of the revised manuscript. 

Comment 3:

The authors combined Tracking and Comprehensive Cohort participants into a single analytical sample. However, did the authors conduct a preliminary assessment to verify the validity of joining the cohorts? This issue is most important for the cognitive tests, which can be influenced by the mode of administration (in-person versus telephone). The authors should check whether the distribution of cognitive test scores was similar across both cohorts and report this information in an appendix.

Authors:

Thank you. The reviewer raised a valid point. We agree that the mode of data collection for cognitive tests was different in tracking and comprehensive cohorts i.e. telephone vs in person respectively. Combining the cohorts in a single analytic sample could raise a possibility of bias in our study results. To address this issue we controlled for ‘type of study cohort’ as a potential confounder in our analysis. This approach has been previous utilized by A. Stinchcombe and colleagues in their recent paper (ref#59). We created a new binary variable ‘type of study cohort (1=tracking, 2=comprehensive) to represent the cohort membership. All multivariate logistic regression were again executed by adding this new variable (type of study cohort) in all models. The new study results (remained the same), based on new regression models, were reported in the revised manuscript. We have added the relevant information of this new variable (type of study cohort) in the section of potential predictors, added in the table 2 for descriptive statistics, and added in the foot notes of Table 4 of the revised manuscript. Updated Table 4 is added to the result section of the revised manuscript.

Comment 4:

The authors should specify that they conducted a complete case analysis. The authors should also examine the potential impact of missing data on their results. For example, were the mean z-scores lower for persons with missing shift work data compared to those with complete shift work data? Was the proportion of people reporting shift work different among participants with missing versus complete cognition data? These issues should be addressed in the analysis to provide some indication as to whether missing data may have biased the results.

Authors:

Thank you. The reviewer raised an important point. We compared the baseline characteristics of participants excluded due to missing information related to shift work schedules and cognitive impairment. We added and clearly elaborated this information in the analysis section, result section and discussion section of the revised manuscript. Details are available in S1 Appendix.

Comment 5:

The CLSA is subject to recruitment bias on cognition because persons with overt cognitive impairment were excluded from the study. Additionally, volunteer bias may also be an issue, as the most cognitively healthy subgroup of the population may have chosen to participate. This is probably why the overall proportion of participants with cognitive impairment decreases as age group increases (Table 2). The authors should discuss how the recruitment of a cognitively healthy sample might bias their results (toward or away from the null).

Authors:

Thank you. The reviewer raised interesting point. We agree that the CLSA data excluded participants with overt cognitive impairment and this raises a possibility of bias. We added and clearly explained this as a potential limitation in the discussion section of the revised manuscript.

Comment 6 :

The data in CLSA did not permit the authors to assess the length of exposure to shift work. Please explain how this limitation may have biased the findings.

Authors:

Thank you for pointing this out. We were not able to assess the length of exposure to shift work and cognitive impairment. The lack of this information is a limitation and suggest potential area for future investigation. This has been added as a potential limitation in the discussion section of the revised manuscript.

Comment 7:

To encourage transparency in the reporting of research results, I strongly recommend the authors report their study in line with the STROBE checklist (https://www.strobe-statement.org/checklists/) and include a copy of the completed checklist as an appendix.

Authors: Thank you. We have provided complete STROBE checklist as an appendix. 

Comment 8:

p. 4, lines 74-75: Comprehensive Cohort participants also provided data during a site visit at their local data collection site. In fact, the entire neuropsychological battery was administered to Comprehensive participants at the data collection site.

Authors:

Thank you for pointing this out. We have added this information in the methods section of the revised manuscript.

Comment 9:

p. 7, line 136: please cite your justifications for including the specific set of covariates described in ‘potential predictors’. Provide the citations that led you to choose these covariates over others.

Authors:

Thank you. We have added the citations that led us to choose these covariates that were utilized in previous studies. 

Comment 10:

p. 11, line 180: change ‘high odds’ to ‘higher odds’.

Authors Thank you for pointing this out. We have corrected this error in the revised manuscript. 

Comment 11:

p. 15, line 247: Please verify whether reference # 55 should actually be reference # 21.

Authors:

Thank you for pointing this out. We have corrected this error in the revised manuscript. In the revised manuscript the study reference is #36. 

Comment 12:

The CLSA-mandated acknowledgement is not complete. Please check with CLSA and obtain the complete acknowledgement.

Authors:

Thank you for pointing this out. We have added complete acknowledgement statement from CLSA to the revised manuscript. 

Reviewer #2

Comment 1:Abstract:

The methods section generically refers to cognitive tests. I think it would be appropriate to make explicit which tests were used.

Authors:

Thank you. We have mentioned the names of all the cognitive tests included in the study, in the method section of the abstract of the revised manuscript.

Comment 2:

Pag. 3, lines 55-56: “A variety of negative health outcomes have been associated with SW, particularly night and rotating SW”. I think readers would be interested in a short, concise but comprehensive list of what adverse effects we are talking about;

Authors:

Thank you. The reviewer makes an important point and we have added and clearly explained the relevant details of all listed adverse effects related to shift work in the introduction section of the revised manuscript. We agree that readers would benefit from this detailed information.

Comment 3:

Pag 3, lines 56-57: “The existing body of literature supports the notion that SW plays a critical role in cognitive functions.”. In my opinion this sentence should be reformulated for clarity… maybe the authors meant “… in cognitive function impairment”;

Authors:

Thank you for pointing this out. We have rephrased the sentence for clarity. The new sentence is “The existing body of literature supports the notion that SW plays a critical role in cognitive function impairment.” 

Comment 4:

I suggest to include in the “Discussion” section a brief paragraph in which the authors discuss how the aforementioned SW characteristics could affect the cognitive functions of exposed workers. In addition, to provide a more comprehensive overview, it might be useful to add a few sentences about what other occupational risk factors (besides shift work) might have adverse effects on the cognitive functions of exposed workers (psychosocial risk factors? heavy metals? solvents? other?)

Authors:

Thank you. We have added more comprehensive details regarding underlying pathophysiological mechanisms linking shift work and negative health outcomes, including cognitive impairment, to the introduction section of the revised manuscript. Also, we have elaborated the mechanisms again in discussion section of the revised manuscript with more focus on mechanisms linking shift work and cognitive impairment. 

In addition, to provide a more comprehensive overview, we added one paragraph about other occupational risk factors (besides shift work) that might have adverse effects on the cognitive functions of exposed workers. We have added and clearly explained this information in the discussion section of the revised manuscript.

Comment 5:

Page 9, Table 2, last column heading: Please correct “impairment”;

Authors:

Thank you for pointing this out. We have corrected this error.

Comment 6:

Page 13, line 197: “…to be exposed to;”. Please delete the semicolon;

Authors:

Thank you for pointing this out. We have corrected this error and deleted the semicolon in revised manuscript.

Comment 7:

Page 13, line 207: “…who reported;”. Please delete the semicolon;

Authors:

Thank you for pointing this out. We have corrected this error and deleted the semicolon in revised manuscript.

---

## [Decision Letter · Decision Letter 1]

24 Jul 2023

The association between shift work exposure and cognitive impairment among middle-aged and older adults: results from the Canadian Longitudinal Study on Aging (CLSA)

PONE-D-23-11981R1

Dear Dr. Khan,

We’re pleased to inform you that your manuscript has been judged scientifically suitable for publication and will be formally accepted for publication once it meets all outstanding technical requirements.

Kind regards,

Mario Ulises Pérez-Zepeda, M.D., Ph.D.

Academic Editor

PLOS ONE

Additional Editor Comments (optional):

Both reviewers state that their previous concerns were satisfactory, therefore I am happy to accept this work. 

Reviewers' comments:

Reviewer's Responses to Questions

**Comments to the Author**

1. If the authors have adequately addressed your comments raised in a previous round of review and you feel that this manuscript is now acceptable for publication, you may indicate that here to bypass the “Comments to the Author” section, enter your conflict of interest statement in the “Confidential to Editor” section, and submit your "Accept" recommendation.

Reviewer #1: All comments have been addressed

Reviewer #2: All comments have been addressed

2. Is the manuscript technically sound, and do the data support the conclusions?

Reviewer #1: Yes

Reviewer #2: Yes

3. Has the statistical analysis been performed appropriately and rigorously? 

Reviewer #1: Yes

Reviewer #2: Yes

4. Have the authors made all data underlying the findings in their manuscript fully available?

Reviewer #1: Yes

Reviewer #2: Yes

5. Is the manuscript presented in an intelligible fashion and written in standard English?

Reviewer #1: Yes

Reviewer #2: Yes

6. Review Comments to the Author

Reviewer #1: The authors have satisfactorily addressed all my comments and the manuscript is suitable for publication.

Reviewer #2: The authors responded satisfactorily to all comments made in the previous review round. The revised version of the manuscript has improved and, in my opinion, can be published in its current form.

7. PLOS authors have the option to publish the peer review history of their article (what does this mean?). If published, this will include your full peer review and any attached files.

Reviewer #1: No

Reviewer #2: No

---

## [Editor Report · Acceptance letter]

28 Jul 2023

PONE-D-23-11981R1 

The association between shift work exposure and cognitive impairment among middle-aged and older adults: results from the Canadian Longitudinal Study on Aging (CLSA) 

Dear Dr. Khan:

I'm pleased to inform you that your manuscript has been deemed suitable for publication in PLOS ONE. Congratulations! Your manuscript is now with our production department. 

Kind regards, 

on behalf of

Dr. Mario Ulises Pérez-Zepeda 

Academic Editor

PLOS ONE